# Genome-wide association meta-analysis identifies pleiotropic risk loci for aerodigestive squamous cell cancers

Corina Lesseur[1,2], Aida Ferreiro-Iglesias[1], James D. McKay[3], Yohan Bossé[4], Mattias Johansson[1], Valerie Gaborieau[1], Maria Teresa Landi[5], David C. Christiani[6], Neil C. Caporaso[5], Stig E. Bojesen[7], Christopher I. Amos[8], Sanjay Shete[9], Geoffrey Liu[10], Gadi Rennert[11], Demetrius Albanes[5], Melinda C. Aldrich[12], Adonina Tardon[13], Chu Chen[14], Liloglou Triantafillos[15], John K. Field[15], Marion Dawn Teare[16], Lambertus A. Kiemeney[17], Brenda Diergaarde[18,19,20], Robert L. Ferris[20], Shanbeh Zienolddiny[21], Stephen Lam[22], Andrew F. Olshan[23], Mark C. Weissler[24], Martin Lacko[25], Angela Risch[26,27,28], Heike Bickeböller[29], Andy R. Ness[30,31], Steve Thomas[31], Loic Le Marchand[32], Matthew B. Schabath[33], Victor Wünsch-Filho[34], Eloiza H. Tajara[35], Angeline S. Andrew[36], Gary M. Clifford[37], Philip Lazarus[38], Kjell Grankvist[39], Mikael Johansson[40], Susanne Arnold[41], Olle Melander[42,43], Hans Brunnström[44], Stefania Boccia[45,46], Gabriella Cadoni[47,48], Wim Timens[49,50], Ma'en Obeidat[51], Xiangjun Xiao[8], Richard S. Houlston[52], Rayjean J. Hung[53]*, Paul Brennan[1]*

**Data Availability Statement:** Genotype data have been deposited dbGaP accession number phs001202.v1.p1 for the oral and pharyngeal study [17] and for the lung data [16] accession numbers phs001273.v3.p2 and phs000876.v2.p1. The

1 Section of Genetics, Genetic Epidemiology Group, International Agency for Research on Cancer, World Health Organization, Lyon, France, 2 Department of Environmental Medicine and Public Health, Icahn School of Medicine at Mount Sinai, New York, New York, United States of America, 3 Section of Genetics, Genetic Cancer Susceptibility Group, International Agency for Research on Cancer, World Health Organization, Lyon, France, 4 Department of Molecular Medicine, Institut universitaire de cardiologie et de pneumologie de Québec, Laval University, Quebec City, Canada, 5 Division of Cancer Epidemiology and Genetics, National Cancer Institute, National Institutes of Health, Bethesda, Maryland, United States of America, 6 Department of Environmental Health, Harvard TH Chan School of Public Health, Massachusetts General Hospital, Boston, Massachusetts, United States of America, 7 Copenhagen General Population Study, Herlev and Gentofte Hospital, Copenhagen, Denmark, 8 Department of Medicine, Baylor college of Medicine, Houston, Texas, United States of America, 9 Department of Biostatistics, The University of Texas M.D. Anderson Cancer Center, Houston, Texas, United States of America, 10 Lunenfeld-Tanenbaum Research Institute of Sinai Health System, University of Toronto, Toronto, Canada, 11 Clalit National Cancer Control Center, Carmel Medical Center and Technion Faculty of Medicine, Haifa, Israel, 12 Department of Thoracic Surgery, Division of Epidemiology, Vanderbilt University Medical Center, Nashville, Tennessee, United States of America, 13 Faculty of Medicine, University of Oviedo and CIBERESP, Oviedo, Spain, 14 Department of Epidemiology, University of Washington School of Public Health and Community Medicine, Seattle, Washington, United States of America, 15 Institute of Translational Medicine, University of Liverpool, Liverpool, United Kingdom, 16 School of Health and Related Research, University Of Sheffield, Sheffield, United Kingdom, 17 Radboud University Medical Center, Nijmegen, The Netherlands, 18 Department of Human Genetics, University of Pittsburgh, Pittsburgh, Pennsylvania, United States of America, 19 Graduate School of Public Health, University of Pittsburgh, Pittsburgh, Pennsylvania, United States of America, 20 UPMC Hillman Cancer Center, University of Pittsburgh, Pittsburgh, Pennsylvania, United States of America, 21 National Institute of Occupational Health, Oslo, Norway, 22 British Columbia Cancer Agency, Vancouver, Canada, 23 Department of Epidemiology, Gillings School of Global Public Health, University of North Carolina at Chapel Hill, Chapel Hill, North Carolina, United States of America, 24 Department of Otolaryngology/Head and Neck Surgery, UNC Lineberger Comprehensive Cancer Center, University of North Carolina at Chapel Hill, Chapel Hill, North Carolina, United States of America, 25 Department of Otorhinolaryngology, Head and Neck Surgery, Maastricht University Medical Center, Maastricht, The Netherlands, 26 University of Salzburg, Department of Biosciences and Cancer Cluster Salzburg, Salzburg, Austria, 27 Division of Epigenomics, DKFZ – German Cancer Research Center, Heidelberg, Germany, 28 Translational Lung Research Center Heidelberg (TLRC-H), Member of the German Center for Lung Research (DZL), Heidelberg, Germany, 29 Department of Genetic Epidemiology, University Medical Center, Georg-August-University Göttingen, Göttingen, Germany, 30 National Institute for Health Research (NIHR) Bristol Biomedical Research Centre, University Hospitals Bristol NHS Foundation Trust, Bristol, United Kingdom, 31 Bristol Dental School, University of Bristol, Bristol, United Kingdom, 32 Epidemiology Program,

summary statistics for the lung squamous dataset are deposited in dbGaP (phs001273.v3.p2). The oral and pharyngeal GWAS summary statistics by cancer site and world region have been deposited in the IEU Open GWAS platform (https://gwas.mrcieu.ac.uk/) under the GWAs IDs: ieu-b-89, ieu-b-90, ieu-b-94, ieu-b-96, ieu-b-93, ieu-b-97, ieu-b-91, ieu-b-95 and 98.

**Funding:** The INTEGRAL-ILCCO OncoArray was supported by the Centre for Inherited Disease Research (26820120008i-0-26800068-1). Genotyping for the oral and oropharyngeal cancer OncoArray was funded through the U.S. National Institute of Dental and Craniofacial Research (NIDCR) grant 1X01HG007780-0. The Integrative Analysis of Lung Cancer Risk and (INTEGRAL) of the International Lung Cancer Consortium (ILCCO) was supported by grants U19-CA148127 and CA148127S1 and more recently by the INTEGRAL grant U19CA203654. ILCCO data harmonization is supported by the Canada Research Chair to R.J.H. and U19 CA203654. C.I.A. is a Research Scholar of the Cancer Prevention Institute of Texas and supported by RR170048. The work of the Houlston Laboratory is funded by Cancer Research UK. The CAPUA study was supported by FIS-FEDER/Spain grant numbers FIS-01/310, FIS-PI03-0365, and FIS-07-BI060604, FICYT/Asturias grant numbers FICYT PB02-67 and FICYT IB09-133, and the University Institute of Oncology (IUOPA), of the University of Oviedo and the Ciber de Epidemiologia y Salud Pública. CIBERESP, SPAIN. The work performed in the CARET study was supported by the National Institute of Health /National Cancer Institute: UM1 CA167462 (PI: Goodman), National Institute of Health U01-CA6367307 (PIs Omen, Goodman); National Institute of Health R01 CA111703 (PI Chen), National Institute of Health 5R01 CA151989-01A1 (PI Doherty). The Liverpool Lung project is supported by the Roy Castle Lung Cancer Foundation. The Harvard Lung Cancer Study was supported by the NIH (National Cancer Institute) grants CA092824, CA090578, CA074386. The Multiethnic Cohort Study was partially supported by NIH Grants CA164973, CA033619, CA63464 and CA148127. The work performed in MSH-PMH study was supported by The Canadian Cancer Society Research Institute (020214), Ontario Institute of Cancer and Cancer Care Ontario Chair Award to R.J.H. and G.L. and the Alan Brown Chair and Lusi Wong Programs at the Princess Margaret Hospital Foundation. The Norway study was supported by Norwegian Cancer Society, Norwegian Research Council. The work in TLC study has been supported in part the James & Esther King Biomedical Research Program (09KN-

University of Hawaii Cancer Center, University of Hawaii, Honolulu, Hawaii, United States of America, **33** Department of Cancer Epidemiology, H. Lee Moffitt Cancer Center and Research Institute, Tampa, Florida, United States of America, **34** Faculdade de Saúde Pública, Universidade de São Paulo, São Paulo, Brazil, **35** Department of Molecular Biology, School of Medicine of São José do Rio Preto, São José do Rio Preto, Brazil, **36** Biomedical Data Science, Geisel School of Medicine at Dartmouth, Dartmouth College, Hanover, New Hampshire, United States of America, **37** Infections Section, Infections and Cancer Epidemiology Group, International Agency for Research on Cancer, World Health Organization, Lyon, France, **38** Department of Pharmaceutical Sciences, College of Pharmacy, Washington State University, Spokane, Washington, United States of America, **39** Department of Medical Biosciences, Umeå University, Umeå, Sweden, **40** Department of Radiation Sciences, Umeå University, Umeå, Sweden, **41** Markey Cancer Center, University of Kentucky, Lexington, Kentucky, United States of America, **42** Department of Clinical Sciences Malmö, Lund University, Malmö, Sweden, **43** Department of Internal Medicine, Skåne University Hospital, Malmö, Sweden, **44** Department of Clinical Sciences, Lund University, Lund, Sweden, **45** Section of Hygiene, University Department of Life Sciences and Public Health, Università Cattolica del Sacro Cuore, Roma, Italia, **46** Department of Woman and Child Health and Public Health - Public Health Area, Fondazione Policlinico Universitario A. Gemelli IRCCS, Roma, Italia, **47** Dipartimento Patologia Testa Collo e Organi di Senso, Istituto di Clinica Otorinolaringoiatrica, Università Cattolica del Sacro Cuore, Roma, Italia, **48** Dipartimento di Scienze dell'Invecchiamento, Neurologiche, Ortopediche e della Testa-Collo, Fondazione Policlinico Universitario A. Gemelli IRCCS, Roma, Italia, **49** Department of Pathology and Medical Biology, University Medical Center Groningen, University of Groningen, Groningen, The Netherlands, **50** GRIAC Research Institute, University of Groningen, Groningen, The Netherlands, **51** Centre for Heart Lung Innovation, St Paul's Hospital, The University of British Columbia, Vancouver, Canada, **52** Division of Genetics and Epidemiology, The Institute of Cancer Research, London, United Kingdom, **53** Prosserman Centre for Population Health Research, Lunenfeld-Tanenbaum Research Institute, Sinai Health System, Toronto, Canada

* rayjean.hung@lunenfeld.ca (RJH); gep@iarc.fr (PB)

## Abstract

Squamous cell carcinomas (SqCC) of the aerodigestive tract have similar etiological risk factors. Although genetic risk variants for individual cancers have been identified, an agnostic, genome-wide search for shared genetic susceptibility has not been performed. To identify novel and pleotropic SqCC risk variants, we performed a meta-analysis of GWAS data on lung SqCC (LuSqCC), oro/pharyngeal SqCC (OSqCC), laryngeal SqCC (LaSqCC) and esophageal SqCC (ESqCC) cancers, totaling 13,887 cases and 61,961 controls of European ancestry. We identified one novel genome-wide significant ($P_{meta}$<5x10$^{-8}$) aerodigestive SqCC susceptibility loci in the 2q33.1 region (rs56321285, *TMEM273*). Additionally, three previously unknown loci reached suggestive significance ($P_{meta}$<5x10$^{-7}$): 1q32.1 (rs12133735, near *MDM4*), 5q31.2 (rs13181561, *TMEM173*) and 19p13.11 (rs61494113, *ABHD8*). Multiple previously identified loci for aerodigestive SqCC also showed evidence of pleiotropy in at least another SqCC site, these include: 4q23 (*ADH1B*), 6p21.33 (*STK19*), 6p21.32 (*HLA-DQB1*), 9p21.33 (*CDKN2B-AS1*) and 13q13.1(*BRCA2*). Gene-based association and gene set enrichment identified a set of 48 SqCC-related genes rel to DNA damage and epigenetic regulation pathways. Our study highlights the importance of cross-cancer analyses to identify pleiotropic risk loci of histology-related cancers arising at distinct anatomical sites.

15), National Institutes of Health Specialized Programs of Research Excellence (SPORE) Grant (P50 CA119997), and by a Cancer Center Support Grant (CCSG) at the H. Lee Moffitt Cancer Center and Research Institute, an NCI designated Comprehensive Cancer Center (grant number P30-CA76292).The Vanderbilt Lung Cancer Study – BioVU dataset used for the analyses described was obtained from Vanderbilt University Medical Center's BioVU, which is supported by institutional funding, the 1S10RR025141-01 instrumentation award, and by the Vanderbilt CTSA grant UL1TR000445 from NCATS/NIH. Dr. Aldrich was supported by NIH/National Cancer Institute K07CA172294 (PI: Aldrich) and Dr. Bush was supported by NHGRI/NIH U01HG004798 (PI: Crawford). The Copenhagen General Population Study (CGPS) was supported by the Chief Physician Johan Boserup and Lise Boserup Fund, the Danish Medical Research Council and Herlev Hospital. The NELCS study: Grant Number P20RR018787 from the National Center for Research Resources (NCRR), a component of the National Institutes of Health (NIH). The Kentucky Lung Cancer Research Initiative was supported by the Department of Defense [Congressionally Directed Medical Research Program, U.S. Army Medical Research and Materiel Command Program] under award number: 10153006 (W81XWH-11-1-0781). Views and opinions of, and endorsements by the author(s) do not reflect those of the US Army or the Department of Defense. This research was also supported by unrestricted infrastructure funds from the UK Center for Clinical and Translational Science, NIH grant UL1TR000117 and Markey Cancer Center NCI Cancer Center Support Grant (P30CA177558) Shared Resource Facilities: Cancer Research Informatics, Biospecimen and Tissue Procurement, and Biostatistics and Bioinformatics. The M.D. Anderson Cancer Center study was supported in part by grants from the NIH (P50CA070907, R01 CA176568) (to X. Wu), Cancer Prevention & Research Institute of Texas (RP130502) (to X. Wu), and The University of Texas MD Anderson Cancer Center institutional support for the Center for Translational and Public Health Genomics. Head and Neck studies included in the VOYAGER consortium were supported by NIDCR RO1 DE025712-01. The University of Pittsburgh head and neck cancer case–control study is supported by US National Institutes of Health grants P50CA097190 and P30CA047904. The Carolina Head and Neck Cancer Study (CHANCE) was supported by the National Cancer Institute (R01CA90731). The Head and Neck Genome Project (GENCAPO) was supported by the

## Author summary

Squamous cell carcinomas are specific type of cancer that can arise in multiple organs of the aerodigestive tract including the lung, oral cavity, oropharynx, larynx and esophagus. Previous studies have shown that aerodigestive squamous cell carcinomas share common environmental risk factors (tobacco smoking and alcohol intake). Here, we investigate genetic factors involved in the risk of aerodigestive squamous cell carcinomas as a group in a large genetic association study involving 13,887 cancer cases and 61,961 controls. We identified one genome-wide significant region within 2q33.1 and 3 other suggestive regions at 1q32.1, 5q31.2 and 19p13.11. Gene-based analyses also identify a list of SqCC-related genes that are involved in DNA damage response and epigenetic regulation. Our results suggest some overlap in the genetic factors influencing the risk of aerodigestive squamous cell carcinomas in European populations and highlights the importance of cross-cancer studies.

## Introduction

The squamous cell carcinomas (SqCC) of the aerodigestive tract [1], lung squamous cell carcinoma (LuSqCC) and head and neck cancers (HNC, >90% SqCCs) including; oral/pharyngeal SqCC (OSqCC), larynx SqCC (LaSqCC), and esophageal SqCC (ESqCC); are strongly associated with common risk factors such as tobacco smoking, alcohol consumption and human papilloma virus (HPV) infection [2]. Similarly, recent molecular characterization studies across anatomically distinct SqCCs have shown that histology is more important than tissue of origin in defining tumor molecular profiles determined by shared features including somatic mutations, copy number alternations, deregulation of DNA methylation and/or gene expression[2–4].

Along with behavioral risk factors, it is increasingly recognized that inherited factors also play a role in aerodigestive SqCC risk. Previous genome-wide association studies (GWAS) have identified multiple genetic risk variants for individual aerodigestive SqCC types; notably variants in smoking-related genes at 15q25.1 for LuSqCC [5–7] and 4q23 variants in alcohol-related genes for upper aerodigestive tract (UADT) cancers [8]. Importantly, candidate-gene and GWAS studies have previously described rare genetic variants linked to aerodigestive SqCC risk; including variants near *BRCA2* (13q13.1), first identified as a risk factor for ESqCC in Middle Eastern populations [9] and later described to increase risk of LuSqCC [10] and UADT SqCC in Europeans [11]. Similarly, at 22q12.1 another rare missense variant within *CHEK2* (rs17879961, p.Ile157Thr) has been linked to reduced risk of lung and UADT SqCCs [10–13]. Such studies provide evidence of genetic pleiotropy across aerodigestive SqCCs, as these variants exert cross-cancer effects possibly related to similar underlying mechanisms (*i.e.* DNA repair). Furthermore, a recent large-scale genome-wide genetic correlation analysis across six solid tumors (breast, colorectal, head/neck, lung, ovary and prostate cancer), highlighted that the strongest genetic correlation was between lung and head and neck cancers [14].

Collectively aerodigestive SqCC are an important public health issue; not only because as a group are amongst the most common type of solid tumors [2], but also due to the increasing global incidence of HPV-related head and neck SqCCs [15]. Identifying genetic risk loci that can have pleiotropic effects across aerodigestive SqCC sites is important for gaining insight into shared or divergent molecular basis of different tumors. To further examine pleiotropic risk genomic regions across LuSqCC [16], OSqCC [17], LaSqCC [8] and ESqCC [8] and to

Fundação de Amparo à Pesquisa do Estado de São Paulo (FAPESP; grants 04/12054-9 and 10/51168-0). The authors thank all the members of the GENCAPO team. The HN5000 study was funded by the National Institute for Health Research (NIHR) under its Programme Grants for Applied Research scheme (RP-PG-0707-10034); the views expressed in this publication are those of the author(s) and not necessarily those of the NHS, the NIHR or the UK Department of Health. The Toronto study was funded by the Canadian Cancer Society Research Institute (020214) and the National Cancer Institute (U19CA148127) and by the Cancer Care Ontario Research Chair. The Rome Study was supported by the Associazione Italiana per la Ricerca sul Cancro (AIRC). The Alcohol-Related Cancers and Genetic Susceptibility Study in Europe (ARCAGE) was funded by the European Commission's fifth framework programme (QLK1-2001-00182), the Italian Association for Cancer Research, Compagnia di San Paolo/FIRMS, Region Piemonte and Padova University (CPDA057222). The funders did not participate in study design, data collection and analysis, decision to publish or preparation of the manuscript.

**Competing interests:** I have read the journal's policy and the authors of this manuscript have the following competing interests. Dr. Ferris has the following financial disclosures: Aduro Biotech, Inc (consulting); Astra-Zeneca/MedImmune (clinical trial, research funding); Bristol-Myers Squibb (advisory board, clinical trial, research funding); EMD Serono (advisory board); MacroGenics, Inc (advisory board); Merck (advisory board, clinical trial); Novasenta (consulting, stock, research funding); Numab Therapeutics AG (advisory board); Pfizer (advisory board); Sanofi (consultant); Tesaro (research funding) and Zymeworks, Inc (consultant). All other authors have no conflicts to disclose.

identify novel associations not detected in single-cancer analyses, we performed a GWAS meta-analysis combining data from the largest existent GWAS in Europeans for these malignancies.

# Results

## Overview

We performed GWAS meta-analysis on aerodigestive SqCC risk including 13,887 cancer cases and 61,961 non-overlapping controls of European ancestry. The SqCC cases comprised 7,426 LuSqCC, 5,452 OSqCC, part of the OncoArray Consortium [16–18], and additional 693 LaSqCC and 316 ESqCC previously included in a upper aerodigestive cancer GWAS [8] (Table 1). Summary associations statistics were used to perform fixed-effects (F-E) and a subset-based meta-analyses using the ASSET software [19]. This approach allows exploration of all possible subsets of studies to identify the strongest association signal, while accounting for subset search multiple testing, and adjusting standard errors to account for overlapping controls between analyses; partial overlap (N = 2,500) between LuSqCC and OSqCC and complete overlap between the ESqCC and LaSqCC. After quality control steps, 8,468, 885 genetic variants with summary statistics in at least three of the four interrogated SqCC types were used for analyses. The quantile–quantile plot (F-E meta-analysis, S1 Fig) shows little evidence of genomic inflation after correcting for sample size ($\lambda = 1.006$). Loci that reached $P_{meta} < 5 \times 10^{-7}$ were considered noteworthy; meta-analysis results for all SNPs below $P < 5 \times 10^{-5}$ are shown in S1 Table. From noteworthy loci, those not previously reported in the lung [16] or oral/pharyngeal [17] analyses (single-cancer $P > 5 \times 10^{-7}$) were considered as novel SqCC regions. We identified one novel aerodigestive SqCC loci at genome-wide significance (F-E meta-analysis, $P_{meta} < 5 \times 10^{-8}$) within 2q33.1. We detected suggestive associations with SqCC risk at 1q32.1, 5q31.2 and 19p13.11, not detected in previous analyses (Fig 1 and Table 2). Other loci that reached $P_{meta} < 5 \times 10^{-7}$ were considered pleiotropic if these had at least two cancer sites at $P < 5 \times 10^{-4}$ and the same effect direction in all tumor sites. Using these criteria, the loci categorized as pleiotropic (4q23, 6p21.32, 6p21.33, 6p22, 9p21.3 and 13q13.1) included 108 SNPs (S2 Table), the lead SNP (lowest $P_{meta}$) for each of these regions is shown in Table 3. In contrast, other known cancer regions that reached the GWAS threshold (12p13.33, 15q25, 19q13.2) or $P_{meta} < 5 \times 10^{-7}$ (4p14, 9q34.1, 10q24.31, 11q21, 15q15.3) in the SqCC meta-analysis were not pleiotropic (Fig 1). We did not observe additional associations reaching the GWAS threshold or suggestive significance in the ASSET subset-based meta-analysis, indicating that at least for the strongest associations, the effects have consistent direction across the examined aerodigestive SqCC types. For noteworthy SNPs, we performed expression quantitative trait (eQTL) analyses with normal lung tissues from the multicenter Lung Microarray Study (S3 Table). We also query these variants in multiple public genomic annotation databases. The Genotype-Tissue Expression (GTEx) for lung and esophageal eQTLs (S4 Table). ROADMAP and the Encyclopedia of DNA Elements (ENCODE) for epi/genomic annotations (S5 Table). The NHGRI-EBI GWAS Catalog (S6 Table) for disease/phenotype associations and the COSMIC catalogue for cancer somatic mutation information (S7 Table). Lastly, we performed a genome-wide gene-based association analysis (GWGAS) of the SqCC meta-analyses results using MAGMA (Multi-marker Analysis of GenoMic Annotation)[20] (S8 Table). To map individual SNPs to genes we used the Functional Mapping and Annotation (FUMA, S9 Table) [21]. Overlapping genes from these were used to assemble a list of aerodigestive SqCC genes (S10 Table) used in enrichment analyses (S11 Table).

**Table 1. Summary of studies included in the aerodigestive SqCC meta-analysis.**

| Study | Tumor site | Cases | Controls[a] | Array | Imputation panel | Imputation Quality | Number of Variants | Covariates | Ancestry | Publication |
|---|---|---|---|---|---|---|---|---|---|---|
| Lung cancer OncoArray | Lung | 7,426 | 55,630 | Illumina OncoArray | 1000 Genomes v3 | $R^2>0.3$ | 10,439,017 | Age, sex, PCs | European | McKay, Hung et al 2017 |
| Oral and oropharynx cancer OncoArray | Oral and Oropharynx | 5,452 | 5,984 | Illumina OncoArray | HRC | $R^2>0.3$ | 7,542,495 | Age, sex, PCs | European | Lesseur et al 2016 |
|  | Oral | 2,698 |  |  |  |  |  |  |  |  |
|  | Oropharynx | 2,414 |  |  |  |  |  |  |  |  |
|  | Other[b] | 340 |  |  |  |  |  |  |  |  |
| UADT cancer GWAS | Larynx | 693 | 2,847 | Illumina Human-Hap300 | HRC | $R^2>0.3$ | 8,840,446 | Age, sex, PCs | European | McKay et al 2011 |
|  | Esophageal | 316 | 2,847 |  |  |  |  |  |  |  |  |

$R^2$ = imputation quality measure; MAF = minor allele frequency; UADT = upper aerodigestive tract; HRC = Haplotype Reference Consortium panel;

[a] Overlapping controls N = 2,500 (lung and oral/oropharynx) and N = 2847 (larynx and esophageal).

[b] Cases with overlapping oral and oropharyngeal tumors.

## Novel loci with pleiotropic aerodigestive SqCC associations

At 2q33.1, the intronic variant rs56321285[A] within the transmembrane protein 237 (*TMEM237*) gene was associated with reduced risk of aerodigestive SqCC (OR = 0.90, $P_{meta}$ = $6.99 \times 10^{-9}$). This association showed little heterogeneity across cancer sites LuSqCC: OR = 0.92, $P = 2.51 \times 10^{-4}$; OSqCC: OR = 0.89, $P = 2.34 \times 10^{-4}$; LaSqCC: OR = 0.79, $P = 3.83 \times 10^{-3}$; ESqCC: OR = 0.80, $P = 3.56 \times 10^{-2}$ (Fig 2A). rs56321285 is in low linkage disequilibrium (LD) with other variants in the region (S2 Fig) including rs10931936 ($r^2$ = 0.02, 1000 Genomes (1KG), Europeans), the lead SNP of a weaker 2q33.1 association (SqCC $OR_{meta}$ = 1.08, $P_{meta}$ = $1.83 \times 10^{-6}$).

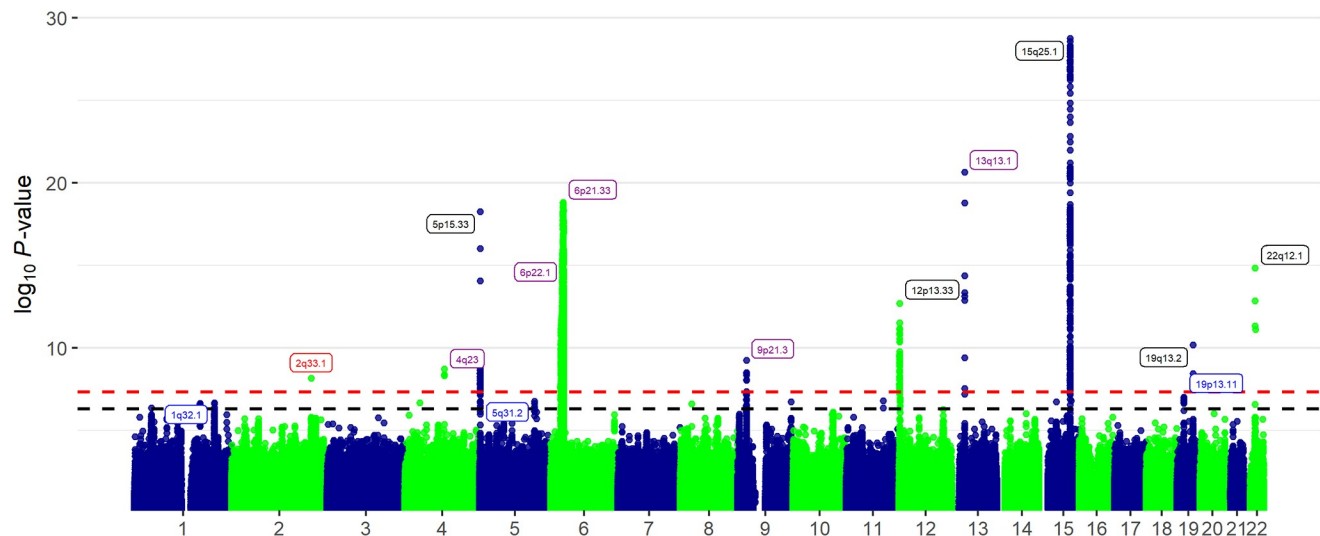

**Fig 1. Manhattan plot of aerodigestive SqCC genome-wide association fixed-effects meta-analysis results.** The y-axis corresponds to −log10 $P$-values, and x-axis to genomic positions. Horizontal red dashed line ($P = 5 \times 10^{-8}$) and black dashed line ($P = 5 \times 10^{-7}$). Highlighted in red are the newly identified pleotropic aerodigestive SqCC loci ($P<5 \times 10^{-8}$), in blue new loci at $P<5 \times 10^{-7}$ and in purple previously identified pleotropic loci (at least 2 cancer sites). Loci labeled in black are loci that reached $P<5 \times 10^{-7}$ but were not pleiotropic associated only in single-cancer analyses.

**Table 2. Novel genomic regions with pleiotropic aerodigestive SqCC associations.**

| Region[a] | EA/OA[b] | Gene | EAF | SqCC site | OR | 95%CI | $P$ | OR$_{meta}$ | 95%CI$_{meta}$ | $P_{meta}$ [c] |
|---|---|---|---|---|---|---|---|---|---|---|
| *Genome-wide significant loci* | | | | | | | | | | |
| 2q33.1 | A/G | *TMEM237* | 0.31 | Lung | 0.92 | 0.88–0.96 | 2.51E-04 | 0.902 | 0.87–0.94 | 6.99E-09 |
| rs56321285 | | | | Oral/oropharynx | 0.89 | 0.83–0.94 | 2.34E-04 | | | |
| 2:202505545 | | | | Larynx | 0.79 | 0.67–0.93 | 3.83E-03 | | | |
| | | | | Esophagus | 0.8 | 0.64–0.98 | 3.56E-02 | | | |
| *Suggestive loci* | | | | | | | | | | |
| 1q32.1 | G/T | *MDM4* | 0.35 | Lung | 1.07 | 1.03–1.11 | 4.63E-04 | 1.08 | 1.05–1.12 | 2.16E-07 |
| rs12133735 | | | | Oral/oropharynx | 1.14 | 1.08–1.21 | 9.82E-06 | | | |
| 1:204556836 | | | | Larynx | 1.05 | 0.92–1.20 | 5.14E-01 | | | |
| | | | | Esophagus | 1 | 0.83–1.19 | 9.74E-01 | | | |
| 5q31.2 | G/A | *TMEM173* | 0.28 | Lung | 1.09 | 1.05–1.14 | 5.29E-05 | 1.09 | 1.05–1.13 | 1.74E-07 |
| rs13181561 | | | | Oral/oropharynx | 1.1 | 1.04–1.17 | 1.74E-03 | | | |
| 5:138850905 | | | | Larynx | 1.12 | 0.96–1.3 | 1.44E-01 | | | |
| | | | | Esophagus | 1.06 | 0.87–1.3 | 5.62E-01 | | | |
| 19p13.11 | A/G | *ABHD8* | 0.3 | Lung | 1.09 | 1.05–1.14 | 2.05E-05 | 1.09 | 1.05–1.12 | 9.86E-08 |
| rs61494113 | | | | Oral/oropharynx | 1.09 | 1.03–1.16 | 4.91E-03 | | | |
| 19:17401859 | | | | Larynx | 1.15 | 1.00–1.31 | 5.22E-02 | | | |
| | | | | Esophagus | 1.05 | 0.87–1.27 | 5.90E-01 | | | |

[a] Lead SNP (lowest $P$, F-E meta-analysis), regions at $P<5 \times 10^{-7}$

[b] EA = Effect allele/OA = other allele; EAF = average effect allele frequency between sites.

[c] F-E meta-analysis (ASSET) accounting for control overlap between Lung SCC and oral/pharynx analysis and larynx and esophageal analysis.

rs10931936 is in LD with nearby *CASP8-ALS2CR12* variants which have been previously linked with risk of multiple cancers in Europeans [22], as well as esophageal and lung cancer in Chinese populations [23,24]. *CASP8* plays an important role in apoptosis; mutations in this gene have been described in 2% of LuSqCC and 6% of UADT SqCC tumors (S7 Table) [25]. However, the 2q33.1 genome-wide significant SNP (rs56321285) associated with aerodigestive SqCC risk, seems independent from *CASP8-ALS2CR12* variants. In eQTL analyses using lung tissues Lung Microarray Study (S3 Table), rs56321285 is a nominally significant cis-eQTL for *AOX2P* (Laval and Groningen datasets) and *CDK15* (Laval and UBC datasets). However, rs56321285 is not a lung or esophageal eQTL in the GTEx catalog [26]. Regulatory annotations from ENCODE [27] and ROADMAP [28] are consistent with rs56321285 mapping to a H3K4me1 enhancer in lung fibroblasts (S5 Table).

The lead SNP (rs12133735) at 1q32.1; the G allele was associated with increased risk of aerodigestive SqCC (OR$_{meta}$ = 1.08, $P_{meta}$ = 2.16x10$^{-7}$, Table 2 and Fig 2B), predominantly driven by the LuSqCC (OR = 1.07, $P$ = 4.63x10$^{-4}$) and OSqCC (OR = 1.14, $P$ = 9.82x10$^{-6}$) results. rs12133735 is located 3' of *MDM4* (S3 Fig) and is a *MDM4* eQTL in all datasets from the lung eQTL study (S3 Table and S4 Fig), and in lung and esophageal GTEx tissues [26] (S4 Table). MDM4 is a crucial negative regulator of p53 and its upregulation has been described as a common p53 inactivation mechanism in tumors [29,30]. In contrast, in our analyses rs12133735-G is associated with lower *MDM4* expression in lung tissues and increased SqCC risk. However, the regulation of *MDM4* expression and interaction with p53 involves complex mechanisms (including alternative splicing) and are reported to differ between normal and cancer tissues [29]. In Europeans, rs12133735[G] is in moderate LD (r$^2$ = 0.61, 1KG) with rs4245739 [C] (SqCC OR$_{meta}$ = 1.07, $P$ = 7.62x10$^{-6}$) which has been associated with increased risk of triple

**Table 3. Variants within known genomic loci with pleiotropic aerodigestive SqCC associations.**

| Region[a] | EA/RA[b] | Gene | EAF | SqCC site | OR | 95%CI | P | OR_meta | 95%CI_meta | P_meta[c] |
|---|---|---|---|---|---|---|---|---|---|---|
| 4q23 | T/C | *ADH1B* | 0.05 | Lung | 0.92 | 0.84–1.01 | 7.35E-02 | 0.80 | 0.74–0.86 | 1.89E-09 |
| rs1229984 | | | | Oral/oropharynx | 0.58 | 0.50–0.67 | 8.32E-13 | | | |
| 4:100239319 | | | | Larynx | 0.67 | 0.49–0.92 | 1.42E-02 | | | |
| | | | | Esophagus | 0.28 | 0.15–0.53 | 9.28E-05 | | | |
| 6p21.33 | G/A | *STK19* | 0.09 | Lung | 1.29 | 1.22–1.37 | 1.41E-16 | 1.26 | 1.19–1.32 | 2.42E-19 |
| rs389884 | | | | Oral/oropharynx | 1.21 | 1.09–1.35 | 4.41E-04 | | | |
| 6:31940897 | | | | Larynx | 1.21 | 0.95–1.52 | 1.19E-01 | | | |
| | | | | Esophagus | 1.24 | 0.92–1.68 | 1.65E-01 | | | |
| 6p21.32 | G/A | *HLA-DQA1* | 0.41 | Lung | 1.17 | 1.12–1.21 | 3.01E-14 | 1.16 | 1.12–1.19 | 4.82E-19 |
| rs9271611 | | | | Oral/oropharynx | 1.15 | 1.08–1.22 | 1.23E-05 | | | |
| 6:32591609 | | | | Larynx | 1.27 | 1.07–1.49 | 5.14E-03 | | | |
| | | | | Esophagus | 1.02 | 0.82–1.28 | 8.36E-01 | | | |
| 9p21.3 | T/C | *CDKN2B-AS1* | 0.30 | Lung | 1.09 | 1.05–1.14 | 1.34E-05 | 1.11 | 1.07–1.14 | 5.55E-10 |
| rs7857345 | | | | Oral/oropharynx | 1.14 | 1.07–1.21 | 3.73E-05 | | | |
| 9:22087473 | | | | Larynx | 1.18 | 1.03–1.36 | 1.91E-02 | | | |
| | | | | Esophagus | 1.07 | 0.89–1.29 | 4.63E-01 | | | |
| 13q13.1 | A/G | *BRCA2* | 0.01 | Lung | 2.12 | 1.77–2.55 | 1.10E-15 | 2 | 1.73–2.32 | 2.30E-21 |
| rs11571815 | | | | Oral/oropharynx | 1.67 | 1.28–2.18 | 1.67E-04 | | | |
| 13:32968550 | | | | Larynx | 3.04 | 1.41–6.57 | 4.53E-03 | | | |
| | | | | Esophagus | 4.73 | 2.14–10.5 | 1.24E-04 | | | |

[a] Lead SNP (lowest *P*, F-E meta-analysis), regions at $P<5\times10^{-7}$;

[b] EA = Effect allele/OA = other allele; EAF = average allele frequency between sites;

[c] F-E meta-analysis (ASSET) accounting for control overlap.

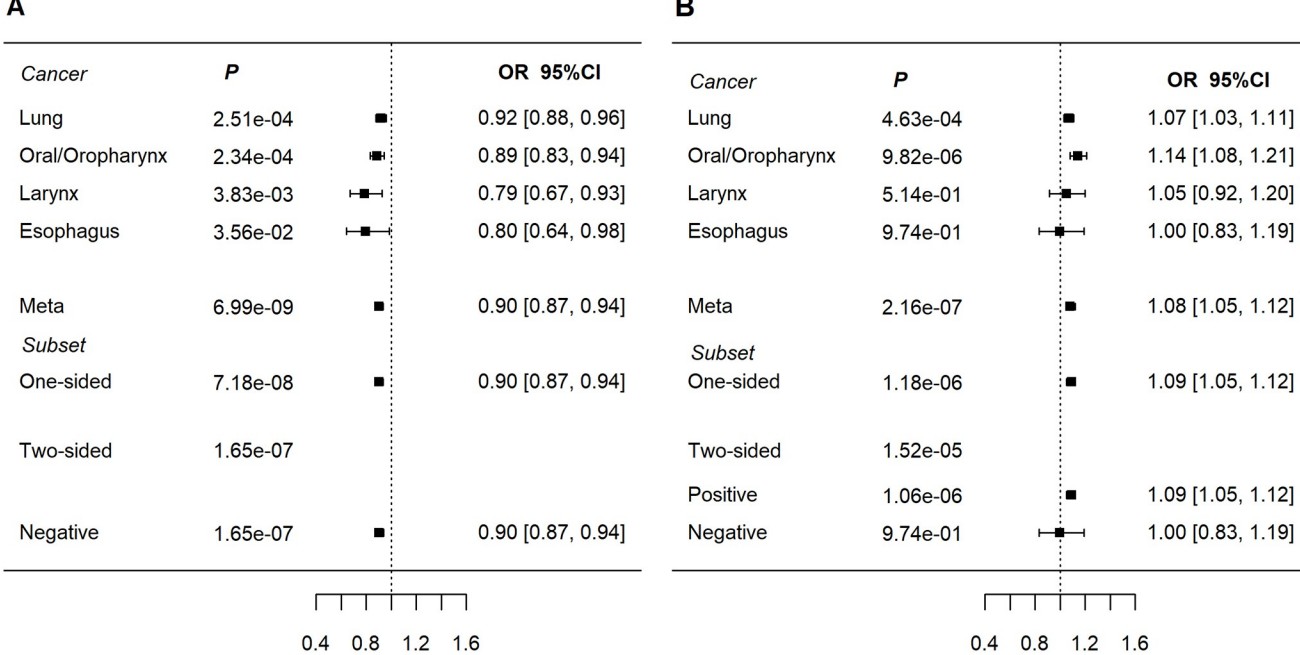

**Fig 2. Forest plots of ORs for new aerodigestive SqCC-related loci.** a) rs56321285 at 2q33.1 (*TMEM23*). Subset refers to subset based meta-analyses (ASSET). b) rs12133735 at 1q32.1 (near MDM4).

negative breast cancer [31,32] and ovarian cancer [33] (S6 Table). Intriguingly, rs4245739 [C] has also been associated with reduced prostate cancer risk (Europeans) [34] and lower risk of all cancers in Asians [35,36]. A candidate-gene study [37] also described associations between risk of HPV16-associated OSqCC and 1q32.1 *MDM4* SNPs including rs11801299 ($r^2 = 0.12$, with rs12133735, 1KG, Europeans), which was marginally associated with SqCC risk in our analysis ($OR_{meta} = 0.91$, $P_{meta} = 1.34 \times 10^{-5}$).

The lead variant at 5q31.2 rs13181561[G] ($OR_{meta} = 1.09$, $P_{meta} = 1.74 \times 10^{-7}$, Fig 3A) near *TMEM173* (S5 Fig) showed homogenous associations across tumor sites but only significant in LuSqCC and OSqCC. rs13181561 is associated with *DNAJC18* and *SPATA24* gene expression in lung tissues (Laval, Groningen and UBC; S3 Table), and of *DNAJC18* in esophagus (GTEx, S4 Table). rs13181561 overlaps with an enhancer in esophageal and lung tissues (S5 Table). Additionally, rs13181561[G] is highly correlated with rs7447927[C] ($r^2 = 0.94$, 1KG, Europeans), the latter ($OR_{meta} = 1.08$, $P = 5.45 \times 10^{-7}$) has been previously linked to increased ESqCC risk in Chinese populations[38] (S6 Table).

Another suggestive SqCC association was detected at rs61494113[A] within 19p13.11; ($OR_{meta} = 1.09$, $P_{meta} = 9.9 \times 10^{-8}$); showed similar odds ratios across SqCCs sites albeit with limited power for larynx and esophageal SqCCs (Table 1 and Fig 3B). Lung eQTL analysis showed rs61494113 as a significant eQTL for *OCEL1* (Laval and Groening, S3 Table). However, the GTEx catalog shows rs61494113 as an esophageal *ABHD8* eQTL and a *BABAM1* splice-QTL (lung and esophagus) but not for *OCEL1* (S4 Table). rs61494113 maps within H3K4me1 histone and DNase marks in normal lung tissues and in lung carcinoma cells (S5 Table). rs61494113[A] is in complete LD with rs56069439[A] ($r^2 = 1$, 1KG, Europeans), also associated with a SqCC risk (S1 Table and S6 Fig) and previously linked with increased risk of ER-negative breast [39] and ovarian [39,40] cancers (S6 Table). The 19p13.11 region of LD contains multiple genes involved in DNA damage repair including *BABAM1* a *BRCA1*-interacting protein[41] and *ANKLE1* [42].

## Known risk loci with pleiotropic aerodigestive SqCC associations

Chromosome 6 showed a large aerodigestive SqCC association signal overlapping the human leukocyte antigen (HLA), region previously identified in the LuSqCC[16] and oral/pharyngeal SqCC[17] cancer analyses. rs389884 near *STK19* was the top pleiotropic SNP at 6p21.33 (SqCC $OR_{meta} = 1.26$; $P = 2.4 \times 10^{-19}$, Table 3 and S2 Table). 6p21.33 SNPs are in moderate LD (rs389884 and rs115785414, r2>0.4, 1KG, Europeans) with variants at 6p22.1, suggesting a common haplotype. We also detected associations at 6p21.32; rs9271611 near *HLA-DQA1* (class II) is not correlated (S7 Fig) with 6p21.33 SNPs, pairwise LD between rs9267123 and rs9271611 $r^2 = 0.09$ (1 KG, Europeans). This second association reduced aerodigestive SqCC risk ($OR_{meta} = 0.85$; $P = 1 \times 10^{-17}$) mainly for OSqCC and LuSqCC (Table 3). These observations are in concert with our previous findings [17,43]; of at least two haplotypes within the HLA region with different effects on cancer risk. Genetic variants at 9p21.33 have also been associated with multiple malignancies including lung adenocarcinoma [16] and OSqCC [17]. rs7857345 (9p21.33 lead SNP mapped to *CDKN2B-AS1*, S8 Fig) was associated with a slight increase in SqCC risk ($OR_{meta} = 1.11$, $P_{meta} = 5.55 \times 10^{-10}$, Table 3). rs7857345 is in LD with rs61271866 ($r^2 = 0.51$) previously associated with ESqCC in Chinese populations [38]. However, rs7857345 is not in LD with rs885518 ($r^2 = 0.006$, 1KG, Europeans), suggesting that this is a different signal to that previously reported for lung adenocarcinoma risk in Europeans[16].

Expectedly, two other previously reported rare variants at 4q23 (rs1229984, *ADH1B*) and 13q13.1 (rs11571815, *BRCA2*) also displayed pleiotropy in this analyses [10–12] [44] (Fig 1, Table 3). Of note, we did not observe pleiotropy for variants at 15q25.1, a known locus related

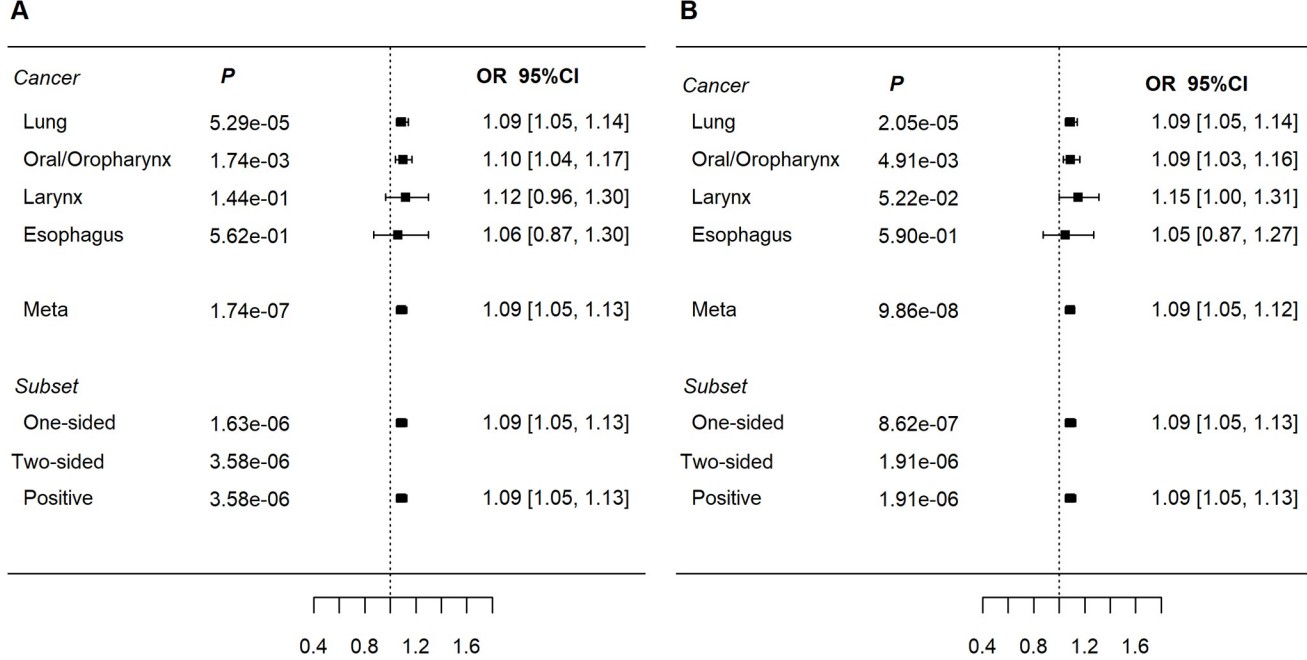

**Fig 3. Forest plots of ORs for new aerodigestive SqCC-related loci.** a) rs13181561 at 5q31.2 (*TMEM173*); b) rs61494113 at 19p13.1 (*ABHD8*). Subset refers to subset based meta-analyses (ASSET).

to lung cancer[5] and smoking behavior [45]. The lead SNP in this region rs55781567 (*CHRNA5*) was prominent for SqCC (OR$_{meta}$ = 1.19; $P_{meta}$ = 1.74x10$^{-29}$), but this result was primarily driven by the LuSqCC (OR = 1.3; $P$ = 4.6x10$^{-41}$) with no effect in any other SqCC site (OSqCC $P$ = 0.26; LaSqCC $P$ = 0.68 and ESqCC $P$ = 0.7, S1 Table and S9 Fig). Variants at 15q25 have been interrogated before in relation to upper aerodigestive cancer risk (including samples used in this study) [46]; significant associations were found only in women and unrelated to smoking behavior suggesting that 15q25.1 SNPs relate differently to LuSqCC compared to oral/oropharyngeal SqCCs. Importantly, none of the published HNC GWAS in Chinese or Europeans have reported associations with 15q25 variants. Thus, while the relation between this locus with smoking behavior and lung cancer is unequivocal; to date there is no evidence of a clear link between 15q25 and head and neck cancer. Future studies including more cases and stratified analyses should examine this further.

## Aerodigestive SqCCs risk genes and pathways

To gain further functional insight into aerodigestive SqCC genetic susceptibility, we used the results of the F-E meta-analyses to map risk variants to genes (FUMA) and to perform a genome-wide gene-based association analysis (GWGAS) using MAGMA. The SNP to gene analyses highlighted 182 genes within 21 genomic regions (S8 Table) and the gene-based analysis identified 51 significant genes related to aerodigestive SqCC (Bonferroni correction $P$<2.67x10-6, S9 Table). Next, we overlapped the results from the two analyses and obtained a list of 48 SqCC-related genes (S10 Table), which includes *TMEM237* (2q33.1), *MDM4* (1q32.1), *AC138517.1* (5q31.2) and *BABAM1* (19p13.11) located within the pleotropic SqCC risk regions identified in the meta-analyses. Expectedly, the HLA region in chromosome 6 had the highest number of genes mapped (24), however and interestingly the most prominent signals were located within 6p22 and included >10 histone genes. Gene-set enrichment analyses using

these SqCC-related genes and canonical pathways resulted in 63 significant gene sets (S11 Table) most of which mapped to DNA damage pathways (telomere, checkpoint, oxidative stress and strand break response) as well as epigenetic regulation pathways related to histones and DNA methylation.

## Discussion

This study identified one novel genome-wide significant loci associated with aerodigestive SqCC risk (2q33.1). Four other loci (1q32.1, 5q31.2 and 19p13.11) showed suggestive associations with SqCC. Amongst known SqCC loci, four showed evidence of pleiotropy across cancer sites. Our results demonstrate the power of cross-cancer analyses of histologically-related tumors to identify genetic risk loci.

It is notable that many of the detected associations are plausibly related to cancer risk. Our results from 2q33.1 and 5q31.2 combined with previous evidence [23,24,38] indicate that these loci relate to SqCC risk across distinct ancestries (Asians and Europeans). Moreover, the signal at 2q33.1 was also proximal to *CASP8-ALS2CR12*, a region previously associated with other cancers [47,48]. Likewise, the 1q32.1 and 19p13.11 genomic regions implicate genes like *MDM4* and *BABAM1*, which have been previously associated with risk of other epithelial malignancies including breast, prostate and ovary [31,36,39,49]. Lastly, these observations not only make our findings more plausible but also expand our understanding of cross-cancer genetic susceptibility and complex biology behind these associations.

The top associations displayed homogenous effect direction across SqCC sites and stronger associations in the F-E meta-analyses compared to the subset-based meta-analyses. This could relate to the shared histology and risk factors of aerodigestive SqCCs. Nonetheless, we cannot rule out heterogeneous SqCC associations that we were not able to detect in our data. However, and not surprisingly, for these loci effect sizes were small (range of $OR_{meta}$ 0.89–1.09) which limited association detection in the smaller single cancer-analysis using commonly applied GWAS *P*-values thresholds. In contrast, most of the known loci that exhibited pleiotropy in our analysis have larger effects sizes, particularly true for the less common variants within *BRCA2* and *CHEK2*.

Our study has several major strengths. Firstly, we leveraged available European GWAS data sets to perform a large-scale meta-analysis of aerodigestive SqCC risk. Secondly, we analyzed tissue-specific gene expression data from multiple studies and integrated these data with publicly available information on epigenetic regulatory profiles of relevant tissues to aerodigestive SqCCs. Thirdly, for the newly discovered loci we also integrated our results with existent data from genetic susceptibility studies in other populations as well as available tumor repository information. However, this study has a number of limitations. The sample sizes for laryngeal and esophageal SqCC were very limited; this impacted our power to identify more signals at the GWAS threshold. The described associations (particularly those at $P > 5x10^{-8}$) could be spurious due to the high testing burden and lack of replication; other studies should examine these regions further to replicate these results. Our criteria to identify pleiotropic loci tried to capture robust loci across multiple aerodigestive SqCC while accommodating for the sample size imbalances across tumor sites. We recognize that this approach did not fully account for multiple testing and could have missed some pleiotropic regions. Pleiotropic studies are limited by sample size of existent GWAS data, as well as the frequency of variants in these regions. Future studies should investigate this further using lager samples, different methodology, and if possible, including SqCCs from other sites (e.g. cervical, anal and bladder). Also, our analyses were restricted to individuals of European ancestry, performing a similar analysis including other genetic backgrounds offers the potential to pinpoint loci that exert effects across

ethnicities. In summary, we provide evidence for one new locus (2q33.1) influencing aerodigestive SqCC risk, and highlight loci for future investigation. Future work should investigate the biological mechanisms underscoring these associations to unearth shared and divergent molecular features of these histologically similar tumors.

## Methods

### Ethics statement

Informed written consent was obtained from all participants, and all contributing studies have been approved by the IARC Institutional Review Board (IRB; references: 14–03, 13–17, 07–02) which requires to obtain local ethics committees approvals prior to their enrolment and evaluation.

### Study population

This meta-analysis includes data from three previous studies of lung squamous cell[16], oral/pharyngeal[17] and upper aerodigestive tract (UADT) cancers[8], totaling 13,887 cases and 61,961 non-overlapping controls. The characteristics and references for each study are summarized in Table 1. The SqCC cases comprise 7,426 LuSqCC, 5,452 OSqCC, part of the OncoArray Consortium (http://epi.grants.cancer.gov/oncoarray/) [16,17], and additional 693 LaSqCC and 316 ESqCC previously included in a upper aerodigestive cancer GWAS[8]. Controls partially overlapped (N = 2,500) between the LuSqCC and OSqCC analyses, and completely overlap (N = 2,847) between the ESqCC and LaSqCC analyses. For this analysis, GWAS summary-statistics for single-site SqCCs were derived using only individuals of European ancestry across multiple epidemiological studies from Europe, North and South America.

### Genotyping and imputation

For each of the studies, genomic DNA samples were previously isolated from blood or buccal cells. Genotyping for the lung and oral/pharyngeal cancers OncoArray Consortium[18] studies, was performed at the Center for Inherited Disease Research (CIDR) using the Illumina OncoArray custom designed for cancer studies. Genotype calls were made in GenomeStudio software (Illumina) using a standardized cluster file for OncoArray studies. The esophageal and larynx cancer cases and controls from the upper aerodigestive tract GWA study[8] were genotyped using the Illumina Sentrix HumanHap300 BeadChip at the Centre d'Etude du Polymorphisme Humain (CEPH) and the Centre National Genotypage (CNG Evry France) as previously described[8]. Genotype data have been deposited dbGaP (https://www.ncbi.nlm.nih.gov/gap/) accession number phs001202.v1.p1 for the oral and pharyngeal study[17] and for the lung data[16] accession numbers phs001273.v3.p2 and phs000876.v2.p1. The lung cancer GWA study[16] was imputed using the 1000 genomes reference panel (phase3) (http://phase3browser.1000genomes.org/index.html/) and the oral/pharyngeal cancer, larynx and esophageal cancer GWAS were imputed using the Haplotype Reference Consortium Panel[50] (http://www.haplotype-reference-consortium.org/) using the University of Michigan Imputation Server [51] (https://imputationserver.sph.umich.edu/). Only variants with imputation quality of R2 > 0.3 were used in the meta-analysis.

### Summary association statistics and meta-analyses

Cancer risk association results from two previous OncoArray Consortia studies (LuSqCC[16] and OSqCC[17]) and the esophageal and laryngeal analyses ORs, P-values and standard errors

for each SNP for each cancer site were obtained using logistic regression with a log additive models adjusted for age, sex and principal components using plink2[52] (https://www.cog-genomics.org/plink2/) and R[53] (http://www.r-project.org/). Summary statistics for the lung SqCC data are deposited in dbGaP (phs001273.v3.p2). The oral and pharyngeal GWAS summary statistics by cancer site and world region have been deposited in the IEU Open GWAS platform (https://gwas.mrcieu.ac.uk/) under the GWAs IDs: ieu-b-89, ieu-b-90, ieu-b-94, ieu-b-96, ieu-b-93, ieu-b-97, ieu-b-91, ieu-b-95 and 98. Meta-analyses were performed using a fixed-effects (F-E) and subset-based meta-analysis using the ASSET software tool [19] (https://dceg.cancer.gov/tools/analysis/asset/). ASSET allows exploration of all possible subsets of studies to identify the strongest association signal, while accounting for subset search multiple testing, and adjust standard errors to account for overlapping controls between analyses; partial overlap (N = 2,500) between the LuSqCC and OSqCC and complete overlap between the ESqCC and LaSqCC analyses. Meta-analysis for a SNP was performed when at least three cancer sites had association results. $P$-values from both analyses were two-sided. Meta-analyses results for fixed-effects and subset-based were considered noteworthy if these reach $P<5x10^{-7}$. Loci were considered as new if these had not been previously reported in the single SqCC cancers analysis ($P>5x10^{-7}$ for any single site). Loci with previously reported LuSqCC or OSqCC were characterized a pleiotropic if: 1) $P_{meta}<5x10^{-7}$; 2) two single cancer association results at $P<5x10^{-4}$ and consistent effect direction across all cancer sites. All analyses were performed using the R statistical environment version 3.4.3[53]. Linkage disequilibrium (LD) calculations ($R^2$) were performed using the LDlink[54] tool and the 1000 Genomes Project European ancestry populations. Regional association plots were generated using stand-alone LocusZoom v1.4[55] (https://github.com/statgen/locuszoom-standalone/).). Forest plots of association results were produce using the metafor R package[56].

## Lung and esophageal cis-eQTLs

To investigate the association between lead SCC variants and mRNA expression, we used three lung eQTL data sets from the Microarray eQTL study. In the Microarray eQTL study [57], lung tissues for eQTL analysis were obtained from patients who underwent lung surgery at three academic sites: Laval University, the University of British Columbia (UBC) and the University of Groningen. Whole-genome gene expression profiling in the lung was performed on a custom Affymetrix array and is available through GEO (https://www.ncbi.nlm.nih.gov/geo/) accession number GSE23546. Genotyping was carried out on the Illumina Human 1M-Duo BeadChip array, data is accessible in dbGaP (phs001745.v1.p1). Genotypes and gene expression levels were available for 408 (Laval University), 342 (Groningen) and 287 (UBC) patients. Microarray and genotypes preprocessing, quality control and eQTL mapping have been described previously[58]. We also investigated top aerodigestive SqCC associations in the public GTEx catalog (V8)[26] for lung and esophageal tissue eQTLs and sQTLs, summary statistics based on RNAseq and genotypes analyses obtained via the GTEx data portal (https://www.gtexportal.org).

## Functional genomic annotation and gene-based analyses

To functionally annotate newly identified aerodigestive SqCC regions, we leveraged multiple resources: the Encyclopedia of DNA Elements (ENCODE)[27] (https://www.encodeproject.org/) and ROADMAP Epigenomics[28] (http://www.roadmapepigenomics.org/) catalogs to obtain epi/genomic regulatory annotations (chromatin states, histones, enhancers, promoters and transcription binding sites) for lung and esophageal tissues and cell-types obtained through HaploReg 4.1 using the HaploR R package[59]); the NHGRI-EBI GWAS Catalog

(v1.0 e98, https://www.ebi.ac.uk/gwas/) [60] for previously reported disease/phenotype associations and the COSMIC catalogue (v90, https://cancer.sanger.ac.uk/cosmic) for cancer somatic mutation information. To provide additional insight into functional and biological mechanisms underlying aerodigestive SqCC genetic susceptibility, we performed a genome-wide gene-based association analysis (GWGAS) of the SqCC meta-analyses results using MAGMA (Multi-marker Analysis of GenoMic Annotation)[20]. We also used the Functional Mapping and Annotation (FUMA, https://fuma.ctglab.nl/)[21] which maps individually significant SNPs to genes. We selected overlapping genes from the MAGMA (Bonferroni-corrected $P$-value $<2.7\text{x}10^{-6}$) and FUMA results were used to assemble a list of genes implicated in aerodigestive SqCC genetic risk. This gene list was used to perform a gene-set analysis for curated canonical biological pathways (containing between 10 and 500 genes) from MSigDB collections[61]; including GO[62], KEGG[63], REACTOME[64] and BIOCARTA[61]. Pathway analyses were performed using MAGMA default settings of 10,000 permutations and applied a Bonferroni correction.

## Supporting information

**S1 Fig. SqCC F-E meta-analyses.** Quantile-quantile plot of the p-values for ASSET F-E meta-analyses results including lung, oral/oropharyngeal, larynx and esophageal SqCCs. (corrected $\lambda = 1.006$).
(TIFF)

**S2 Fig. Regional association plot at 2q33.1.** Chromosome positions (x-axis) and -log10 $P$-value (y-axis) SqCC meta-analysis at 2q33.1. Genetic variants colored red according to their LD with rs56321285 (2q33.1 lead SNP) and colored in blue according to LD values with second lead SNP rs1830298. rs563321285 and rs1830298 $r^2 = 0.02$.
(TIF)

**S3 Fig. Regional association plot at 1q32.1.** Chromosome positions (x-axis) and -log10 $P$-value (y-axis) of SqCC F-E meta-analysis at 1q32.1. Genetic variants are colored according to their LD with the rs12133735 (red) and with rs4245739 (blue) a variant previously associated with cancer risk; rs12133735 and rs4245739 ($r^2 = 0.63$).
(TIF)

**S4 Fig. rs12133735 *MDM4* lung eQTL.** Boxplots for rs12133735 and *MDM4* gene expression in 3 datasets from the Microarray eQTL study, from left to right: Laval University, University of British Columbia (UBC) and 3. University of Groningen.
(TIF)

**S5 Fig. Regional association plot at 5q31.2.** Chromosome positions (x-axis) and -log10 $P$-value (y-axis) SqCC F-E meta-analysis at 5q31.2. Genetic variants colored according to their LD with the labeled SNP (purple diamond). rs13181561 and rs7447927 ($r^2 = 0.94$).
(TIF)

**S6 Fig. Regional association plot at 19p13.11.** Chromosome positions (x-axis) and -log10 $P$-value (y-axis) SqCC F-E meta-analysis at 19p13.11. Genotyped and imputed variants colored according to their LD with the labeled SNP (purple diamond). rs61494113 and rs56069439 $r^2 = 1$.
(TIF)

**S7 Fig. Regional association plot at 6p22.1- 6p21.33.** Chromosome positions (x-axis) and -log10 $P$-value (y-axis) SqCC F-E meta-analysis at 6p22.1- 6p21.33. Variants colored according

to their LD with SNP rs9267123 (lead variant at 6p21.33). rs3116813 (6p22.1) is in moderate LD with rs9267123 ($r^2$ = 0.5). rs1049213 at 6p21.33 is not correlated with rs9267123 ($r^2$ = 0.01).
(TIF)

**S8 Fig. Regional association plot at 9p21.3.** Chromosome positions (x-axis) and -log10 *P*-value (y-axis) SqCC meta-analysis at 9p21.3. Variants colored according to their LD with SNP rs7857345 (9p21.3 lead variant).
(TIF)

**S9 Fig. Regional association plot at 15p25.1.** Regional association plot at 15q25 Chromosome positions (x-axis) and -log10 P-value (y-axis). A. aerodigestive SqCC P-values; B. Lung SqCC P-values; C. Oral and oropharyngeal cancer SqCC P-values. Genetic variants colored red according to their LD with rs55781567 (lowest P-value at 15q5 in the meta-analysis).
(TIF)

**S1 Table. Results with *P*<5x10$^{-5}$ aerodigestive SqCC meta-analyses.** All variants with *P*<5x10$^{-5}$ in the fixed-effects (F-E) ASSET meta-analyses of aerodigestive SqCC. Results for each SqCC site are also shown.
(XLSX)

**S2 Table. Pleiotropic aerodigestive SqCC risk variants.** 108 variants with $P_{meta}$<5x10$^{-7}$; in the fixed-effects (F-E) meta-analyses; two single-cancer analyses at *P*<5x10$^{-4}$ and consistent effect direction across cancer sites.
(CSV)

**S3 Table. Lung Cis-eQTLs for aerodigestive SqCC loci.** Cis-eQTLs for novel SqCC loci in the lung Microarray eQTL study datasets.
(XLSX)

**S4 Table. Cis-eQTLs and cis-sQTLs for new SqCC loci.** Lung and esophageal Cis-eQTLs and cis-sQTLs in the GTEx catalog V8 for new SqCC loci.
(XLSX)

**S5 Table. Chromatin states and histone marks in lung and esophageal tissues or cells for new SqCC loci.** Chromatin and histone annotations for new SqCC loci from the Roadmap and ENCODE projects.
(XLSX)

**S6 Table. Summary of reported cancer risk associations within the newly SqCC risk loci.** NHGRI-EBI Catalog (v1.0 e98 2020-02-08) reported cancer risk associations for lead SNP (or proxies $r^2$>0.6) within the new SqCC loci.
(XLSX)

**S7 Table. Aerodigestive loci genes with somatic mutations.** Genes within SqCC new loci with somatic mutations in the COSMIC catalogue. (release v90, 5th September 2019).
(XLSX)

**S8 Table. Significant results from the gene-based aerodigestive SqCC associations.** Analyses performed with MAGMA with 18669 protein-coding genes.
(XLSX)

**S9 Table. Aerodigestive SqCC results from the FUMA SNPs to genes mapping.**
(XLSX)

**S10 Table. Genes overlapping between FUMA and MAGMA analyses.**
(XLSX)

**S11 Table. Aerodigestive SqCC results from gene set enrichment analyses.**
(XLSX)

## Acknowledgments

We acknowledge all the participants involved in this research and the funders and support.

The authors would like to thank the staff at the Respiratory Health Network Tissue Bank of the FRQS for their valuable assistance with the lung eQTL data set at Laval University. The lung eQTL study at Laval University was supported by the Fondation de l'Institut Universitaire de Cardiologie et de Pneumologie de Québec, the Respiratory Health Network of the FRQS and the Canadian Institutes of Health Research (MOP-123369). Y. Bossé holds a Canada Research Chair in Genomics of Heart and Lung Diseases.

We thank the ARCAGE study investigators and team including: Pagona Lagiou, Tatiana V. Macfarlane, Franco Merletti, Jerry Polesel, Kristina Kjaerheim, Max Robinson, Wolfgang Ahrens, Lorenzo Simonato, Ariana Znaor, Xavier Castellsague (deceased June 2016), David I. Conway, Ivana Holcátová, Claire M. Healy and Peter Thomson. We thank L. Fernandez for her contribution to the IARC ORC multicenter study. We are also grateful to S. Koifman for his contribution to the IARC Latin America multicenter study (S. Koifman passed away in May 2014).

Where authors are identified as personnel of the International Agency for Research on Cancer / World Health Organization, the authors alone are responsible for the views expressed in this article and they do not necessarily represent the decisions, policy or views of the International Agency for Research on Cancer / World Health Organization.

## Author Contributions

**Conceptualization:** Corina Lesseur, Rayjean J. Hung, Paul Brennan.

**Data curation:** Xiangjun Xiao.

**Formal analysis:** Corina Lesseur, Aida Ferreiro-Iglesias, Yohan Bossé, Valerie Gaborieau.

**Funding acquisition:** Christopher I. Amos, Rayjean J. Hung, Paul Brennan.

**Project administration:** Corina Lesseur, Aida Ferreiro-Iglesias.

**Resources:** James D. McKay, Mattias Johansson, Maria Teresa Landi, David C. Christiani, Neil C. Caporaso, Stig E. Bojesen, Christopher I. Amos, Sanjay Shete, Geoffrey Liu, Gadi Rennert, Demetrius Albanes, Melinda C. Aldrich, Adonina Tardon, Chu Chen, Liloglou Triantafillos, John K. Field, Marion Dawn Teare, Lambertus A. Kiemeney, Brenda Diergaarde, Robert L. Ferris, Shanbeh Zienolddiny, Stephen Lam, Andrew F. Olshan, Mark C. Weissler, Martin Lacko, Angela Risch, Heike Bickeböller, Andy R. Ness, Steve Thomas, Loic Le Marchand, Matthew B. Schabath, Victor Wünsch-Filho, Eloiza H. Tajara, Angeline S. Andrew, Gary M. Clifford, Philip Lazarus, Kjell Grankvist, Mikael Johansson, Susanne Arnold, Olle Melander, Hans Brunnström, Stefania Boccia, Gabriella Cadoni, Wim Timens, Ma'en Obeidat, Richard S. Houlston, Rayjean J. Hung, Paul Brennan.

**Supervision:** Rayjean J. Hung, Paul Brennan.

**Writing – original draft:** Corina Lesseur, Aida Ferreiro-Iglesias, Rayjean J. Hung, Paul Brennan.

**Writing – review & editing:** James D. McKay, Mattias Johansson, Maria Teresa Landi, David C. Christiani, Neil C. Caporaso, Stig E. Bojesen, Christopher I. Amos, Sanjay Shete, Geoffrey Liu, Gadi Rennert, Demetrius Albanes, Melinda C. Aldrich, Adonina Tardon, Chu Chen, Liloglou Triantafillos, John K. Field, Marion Dawn Teare, Lambertus A. Kiemeney, Brenda Diergaarde, Robert L. Ferris, Shanbeh Zienolddiny, Stephen Lam, Andrew F. Olshan, Mark C. Weissler, Martin Lacko, Angela Risch, Heike Bickeböller, Andy R. Ness, Steve Thomas, Loic Le Marchand, Matthew B. Schabath, Victor Wünsch-Filho, Eloiza H. Tajara, Angeline S. Andrew, Gary M. Clifford, Philip Lazarus, Kjell Grankvist, Mikael Johansson, Susanne Arnold, Olle Melander, Hans Brunnström, Stefania Boccia, Gabriella Cadoni, Wim Timens, Ma'en Obeidat, Richard S. Houlston, Rayjean J. Hung, Paul Brennan.

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
