## [Decision Letter · Decision Letter 0]

17 Jul 2020

Dear Dr Brennan,

Thank you very much for submitting your Research Article entitled 'Genome-wide association meta-analysis identifies pleiotropic risk loci for aerodigestive squamous cell cancers' to PLOS Genetics. Your manuscript was fully evaluated at the editorial level and by independent peer reviewers. The reviewers appreciated the attention to an important problem, but raised some substantial concerns about the current manuscript. Based on the reviews, we will not be able to accept this version of the manuscript, but we would be willing to review again a much-revised version. We cannot, of course, promise publication at that time.

If you decide to revise the manuscript for further consideration at PLOS Genetics, please aim to resubmit within the next 60 days, unless it will take extra time to address the concerns of the reviewers, in which case we would appreciate an expected resubmission date by email to plosgenetics@plos.org.

[LINK]

We are sorry that we cannot be more positive about your manuscript at this stage. Please do not hesitate to contact us if you have any concerns or questions.

Yours sincerely,

Stephen J. Chanock

Guest Editor

PLOS Genetics

Peter McKinnon

Section Editor: Cancer Genetics

PLOS Genetics

The editors would be willing to consider a new manuscript in which the current work serves as a starting point, and that includes a more thorough and rigorous presentation of the statistical rigor, including the rationale and interpretation.

There was a substantial difference of opinion between reviewers about the statistical analyses and interpretation.

A major revision should address these issues.

Reviewer's Responses to Questions

**Comments to the Authors:**

Reviewer #1: This manuscript presents a meta-analysis of four aerodigestive squamous cell carcinoma (SqCC) GWAS. The study primarily searches for new loci that reach significance via meta-analysis and for evidence of pleiotropy at previously-identified loci (i.e., effects on two or more cancer subtypes). While power is limited, the analysis looks generally well-done. The authors find one new genome-wide significant locus (TMEM237) and three new loci with suggestive significance (P<5e-7); of the three suggestive loci, the MDM4 locus seems quite plausible. However, I have a concern about statistical significance in the pleiotropy analysis and have a few other questions and comments.

1. In their pleiotropy analyses, the authors used a significance threshold of P<5e-3 in at least one other cancer site, but they do not provide justification for this threshold. I am concerned that P<5e-3 does not sufficiently correct for multiple hypotheses tested, which at least include 10 loci x 3 other cancer sites. Also, was only one SNP tested per locus in these pleiotropy analyses, or were multiple SNPs tested? Line 343 "The lead SNP exhibiting pleiotropy..." suggests the latter; if so, then even more hypotheses were tested.

2. Table 2 is titled "Novel genomic regions..." but only the first of the four listed associations is actually genome-wide significant. The authors should separate significant vs. suggestive results in the table. They could also consider moving some of the main text describing suggestive results (which is quite long) to a supplementary note.

3. The overview of analyses indicates that the pleiotropy analyses considered loci that previously reached P<5e-7 in a single-cancer analysis, but the 9p21.3 locus in Table 3 does not appear to satisfy this criterion. Perhaps they should just use a criterion of loci reaching genome-wide significance in their meta-analysis?

4. The authors note that their lead SNP at the MDM4 locus is an eQTL for MDM4 and that MDM4 upregulation can inactivate p53, leading to cancer. Is the eQTL effect direction consistent with this proposed mechanism?

5. A few software packages mentioned in the main text (e.g., ASSET and FUMA) are not defined until Methods. Providing brief descriptions in the main text would improve readability.

6. The legend of Figure 1 indicates that the y-axis was truncated at P=5e-30. Indicating the most significant P-value would be helpful (either in the figure or just by stating it in the legend).

Reviewer #2: This paper combines the results from genome-wide association studies (GWAS) of four aerodigestive squamous cell cancers in an effort to identify (a) novel loci that area associated with risk for more than one of the four cancers and (b) identify regions (perhaps already identified by one or more GWAS) that are plausibly associated with more than one cancer. The authors adopt two sensible and complementary approaches to combine results (i) a simple fixed-effect meta analysis (which will have greatest power when an alleles is associated with all four cancers in the same direction and magnitide) and (ii) a subset-based test (which allows for the possibility that not all cancers will be associated with an allele or in the same direction). They report one new locus and a handful of pleiotropic loci.

The paper's methods are reasonable and sound, and the results are reported with care and appropriate caution (given the quite small sample sizes for two of the studied cancers). Like many GWAS, the ultimate biological implications of the findings are unclear and left for others to sort out--not that there's anything wrong with that. My one substantive comment has to do with the striking lack of detectable pleiotropy at 15q25.1. Considering that lung cancer and head and neck cancer both have very strong genetic correlations with smoking behaviors (see ref 14), it's quite puzzling that the locus with the strongest cigarettes per day association is associated with LuSqCC but not OSqCC. This really deserves more comment--why do the authors think this is the case? Are the analyses from ref 14 not broken out by subtype, or at least not the the same way as here?

My remaining comments are minor presentation and grammar points. One more good read over by a copyeditor to clean up run-on sentences, subject-verb agreement and the like would be helpful.

line 217 "...fixed-effects (F-E) ^{meta-analysis} and a subset-based meta-analysis ^{using ASSET}." To disambiguate.

lines 220-221 "Loci that reached suggestive .... considered noteworthy." Circular definitions. "Colors we defined as red were labeled as crimson." Instead: "Loci with p<5e-7 were considered noteworthy" or "We defined suggestive genome-wide significance as p<5e-7."

line 225 Can you give some motivation as to why p<5e-3 was considered as evidence for pleiotropic effects on a trait at a locus known to be genome-wide significantly associated with another trait?

lines 226-228. "Additionally... 19p13.11." This is not a sentence. Delete "that"?

lines 273-276. Not a sentence. Seems to be two sentences pasted together, with the second sentence missing a verb.

line 306 "this analysis" not "this analyses"

line 333 state the reference panel used to calculate r2. Obviously the r2 cannot be 1 in the study samples or the p-values for rs614etc and rs560etc would be identical.

lines 344-346 and throughout the text (tables are better about this): SNPs are not associated with increases or decreases in risk, alleles (relative to other alleles) are. If you must use "[xxx] is associated with increased risk" language, you have to refer to the effect allele. "rs12345[G] is associated with an increased risk," not "rs12345 is associated with an increased risk."

line 352 see comment on 344-346.

line 358 see comment on 344-346.

line 365 see comment on 344-346.

line 430 no comma after based

lines 442-443 I don't know how "interesting" this observation is. You see what you are powered to see. There may be lots of pleiotropic loci with heterogeneous effects, but in these sample sizes, they may be hard to detect--certainly less power than the SNPs where everything aligns. This sentence is a little tautological, a little post-hoc-power-calculation-y--"we were well powered to detect the things we detected." You should note that there may be heterogeneous pleiotropic loci that you've missed b/c of low power.

line 455 discovered not discover

lines 457-459 Run on. Semi-colon or full stop after "limited."

Reviewer #3: This manuscript describes an analysis in which GWAS results from multiple from multiple aero-digestive squamous cell cancers (lung, oral cavity, oropharynx, larynx, and esophagus) are combined in order to identify loci with pleiotropic effects on multiple cancer types. This is a novel and unexplored hypothesis. There is one novel association observed at a genome-wide significance P-value threshold (in the TMEM237 gene region). All other loci passing this threshold have been reported previously for specific cancer types. The authors report several suggestive association signals and describe pleiotropy for previously identified association signals.

My primary concerns is the lack of replication for the novel locus and the strong emphasis on describing and characterizing the suggestive association signals (which seems like more attention that is warranted based on the statistical evidence). A few additional comments are below:

Introduction: On point of clarification: this GWAS is Europeans only? Or does it also include individuals of European ancestry from outside of Europe?

In my view, there is an over-emphasis of “suggestive” association signals in this paper (i.e., associations with P<10-7). Several signals passing this threshold are expected under the null. A substantial amount of text is this paper is devoted to discussing the biological and epidemiological evidence supporting regions identified at this suggestive P-value threshold.

A P-value threshold of 5x10-3 was used to identify pleiotropic loci. How was this threshold determined? Consider a systematic approach to test for pleiotropy with an explicit multiple testing adjustment.

**Have all data underlying the figures and results presented in the manuscript been provided?**

Reviewer #1: Yes

Reviewer #2: **No: **No mention is made re: availability of the summary statistics from the four GWAS contributing to this analysis. To the best of my knowledge, none are publicly available.

Reviewer #3: Yes

PLOS authors have the option to publish the peer review history of their article (what does this mean?). If published, this will include your full peer review and any attached files.

Reviewer #1: No

Reviewer #2: No

Reviewer #3: No

---

## [Decision Letter · Decision Letter 1]

12 Oct 2020

Dear Dr Brennan,

Thank you very much for submitting your Research Article entitled 'Genome-wide association meta-analysis identifies pleiotropic risk loci for aerodigestive squamous cell cancers' to PLOS Genetics. Your manuscript was fully evaluated at the editorial level and by independent peer reviewers. The reviewers appreciated the attention to an important topic but identified some aspects of the manuscript that should be improved.

We therefore ask you to modify the manuscript according to the review recommendations before we can consider your manuscript for acceptance. Your revisions should address the specific points made by each reviewer.

[LINK]

Yours sincerely,

Stephen J. Chanock

Guest Editor

PLOS Genetics

Peter McKinnon

Section Editor: Cancer Genetics

PLOS Genetics

The authors have answered nearly all of the queries and shoudl provide the details of exactly how the summary data can be available for verification and further exploration.

Reviewer's Responses to Questions

**Comments to the Authors:**

Reviewer #1: The authors have satisfactorily addressed my comments. Providing some justification for the significance criterion of "two single cancer association results at P<5x10-4" would still be helpful. (The response letter seems to have a typo and states P<0.0001 instead of P<5x10-4.)

Reviewer #2: The authors have thoughtfully responded to my comments.

Reviewer #3: The authors have addressed my comments.

**Have all data underlying the figures and results presented in the manuscript been provided?**

Reviewer #1: Yes

Reviewer #2: **No: **It is not clear that summary statistics for individual cancers or the cross-cancer analyses have been made publicly available. Hence it would be nearly impossible to reproduce the Manhattan plot--researchers would have to download dbGAP data sets, run their own QC and GWAS analyses, then run their own cross-cancer analyses. Inevitably, small differences in analysis pipelines will lead to different results.

Reviewer #3: Yes

PLOS authors have the option to publish the peer review history of their article (what does this mean?). If published, this will include your full peer review and any attached files.

Reviewer #1: No

Reviewer #2: No

Reviewer #3: No

---

## [Editor Report · Decision Letter 2]

5 Nov 2020

Dear Dr Brennan,

We are pleased to inform you that your manuscript entitled "Genome-wide association meta-analysis identifies pleiotropic risk loci for aerodigestive squamous cell cancers" has been editorially accepted for publication in PLOS Genetics. Congratulations!

Yours sincerely,

Stephen J. Chanock

Guest Editor

PLOS Genetics

Peter McKinnon

Section Editor: Cancer Genetics

PLOS Genetics

Comments from the reviewers (if applicable):

**Data Deposition**

http://datadryad.org/submit?journalID=pgenetics&manu=PGENETICS-D-20-00903R2

**Press Queries**

---

## [Editor Report · Acceptance letter]

28 Feb 2021

PGENETICS-D-20-00903R2 

Genome-wide association meta-analysis identifies pleiotropic risk loci for aerodigestive squamous cell cancers 

Dear Dr Brennan, 

We are pleased to inform you that your manuscript entitled "Genome-wide association meta-analysis identifies pleiotropic risk loci for aerodigestive squamous cell cancers" has been formally accepted for publication in PLOS Genetics! Your manuscript is now with our production department and you will be notified of the publication date in due course.

With kind regards,

Alice Ellingham

PLOS Genetics

On behalf of:
